

# *In situ* nuclear magnetic resonance response of permafrost and active layer soil in boreal and tundra ecosystems

M. Andy Kass[1], Trevor P. Irons[2], Burke J. Minsley[1], Neal J. Pastick[3,4], Dana R.N. Brown[5], and Bruce K. Wylie[6]

[1]Crustal Geophysics and Geochemistry Science Center, U.S. Geological Survey, Denver CO 80225, USA
[2]Energy and Geoscience Institute, Department of Civil and Environmental Engineering, University of Utah, Salt Lake City, UT 84112, USA
[3]Stinger Ghaffarian Technologies, Inc., Sioux Falls SD 57198, USA
[4]Department of Forest Resources, University of Minnesota Twin Cities, St. Paul MN 55108, USA
[5]Institute of Arctic Biology, University of Alaska Fairbanks, Fairbanks AK 99775, USA
[6]Earth Resources Observation and Science Center, U.S. Geological Survey, Sioux Falls SD 57198, USA

*Correspondence to:* M.A. Kass (mkass@usgs.gov)

**Abstract.** Characterization of permafrost, particularly warm and near-surface permafrost which can contain significant liquid water, is critical to understanding complex interrelationships with climate change, ecosystems, and disturbances such as wildfires. Understanding the vulnerability and resilience of permafrost requires an interdisciplinary approach, relying on (for example) geophysical investigations, ecological characterization, direct observations, remote sensing, and more. As part of a
multi-year investigation into the impacts of wildfires to permafrost, we have collected *in situ* measurements of the nuclear magnetic resonance (NMR) response of active layer and permafrost in a variety of soil conditions, types, and saturations. In this paper, we summarize the NMR data and present quantitative relationships between active layer and permafrost liquid water content and pore sizes. Through statistical analyses and synthetic freezing simulations, we also demonstrate that borehole NMR can image the nucleation of ice within soil pore spaces.

# 1 Introduction

Permafrost is an integral part of high-latitude ecosystems, underlying nearly 24% of the exposed land area in the Northern Hemisphere (Zhang et al., 1999). Much of this permafrost is considered 'warm' or 'vulnerable,' where the temperature is within 5 degrees of the freezing point. For example, Henry and Smith (2001) estimate that approximately 54% of the permafrost in Canada exists between -5°C and 0°C. This warm permafrost is often discontinuous and quite vulnerable to changes in air
temperature, snow accumulation, and surface disturbance (Osterkamp et al., 2000; Hinzman et al., 2005; Jorgenson et al., 2010).

Permafrost thaw can have substantial influences on terrain and ecosystems, with consequences we are only beginning to understand and quantify. Perhaps most notable is the liberation and mobilization of vast quantities of carbon sequestered in near-surface permafrost. This reservoir can become bio-available, and is estimated to be on the order of 20-60 kg/m$^3$ (Brown
et al., 2003; Zimov et al., 2006). This represents the potential for positive feedbacks to climate warming (Schuur et al., 2015).



Tarnocai et al. (2009) estimate that half of all soil organic carbon exists within permafrost with Hugelius et al. (2014) estimating a slightly lower value. However, equally significant is the interrelationship between boreal forests (taiga) and discontinuous permafrost. These forests compose at least half of all the land area underlain by permafrost, and degradation of ice-rich permafrost can have severe consequences on these ecosystems, resulting in the conversion of forestland into bogs or fens in

lowland environments (Osterkamp et al., 2000; Lara et al., 2016). Similar water impoundment can occur with subsidence in tundra ecosystems. Degradation can fundamentally alter local and regional hydrology, with divergent consequences depending on permafrost ice-content, topography, and soil texture (Jorgenson et al., 2013). Extreme effects such as thermokarst-related subsidence of up to 6 meters (e.g., Yoshikawa and Hinzman, 2003) are possible. Catastrophic events such as wildfires can initiate or exacerbate permafrost degradation by combustion of surface organics, thereby increasing soil heat flux (e.g., Yoshikawa

and Hinzman, 2003; Minsley et al., 2016b). All of these effects lead to increased ecosystem stress through increased vulnerability and altered long-term forest communities. This list is far from complete; we refer the reader to Hinzman et al. (2005) for a comprehensive treatment.

With such a complex relationship between permafrost and the environment–potentially leading to drastic changes in ecosystems in a warming climate–it is clearly critical to understand the characteristics of vulnerable permafrost, especially *in situ*.

Changes to structure and composition of both frozen and unfrozen highly saturated soils with extraction can alter sample results in laboratory tests, and understanding *in situ* characteristics will allow for repeat and continuous monitoring. The hydrology of permafrost is particularly interesting, as significant unfrozen water can remain in permafrost at surprisingly low temperatures, well below what is considered 'warm' permafrost (e.g., Davis, 2001; Kleinberg and Griffin, 2005; Watanabe and Wake, 2009).

Previous studies have successfully investigated water contents in permafrost using nuclear magnetic resonance (NMR) in

drill cores (e.g., Kleinberg and Griffin, 2005) or in constructed substrates (e.g., Watanabe and Mizoguchi, 2002; Sparrman et al., 2004). Attempts at borehole NMR (bNMR) measurement *in situ* have generally been performed in oil field applications, and have failed due to degradation of the permafrost as a consequence of the drilling mud thermal interaction and infiltration (Kleinberg and Griffin, 2005). Recent advances in miniaturization and improvements in instrumentation of borehole NMR tools have allowed measurements to be made in drillholes that do not require drilling mud.

In this paper, we present the results of borehole nuclear magnetic resonance surveys as part of a larger multi-year interdisciplinary study in Alaska (Minsley et al., 2016b) to investigate water content and porosity characteristics of the permafrost and active layer *in situ*, without requiring analysis of core. We observe significant quantities of liquid water within near-surface (upper 3 meters) permafrost and describe the distribution of that water as a function of apparent pore size. The dataset includes data from discontinuous and continuous permafrost in boreal forest and tundra ecosystems, as well as directly within a known

ice wedge. Performing bNMR measurements in both the active layer and in permafrost, as well as in disturbed areas and identical undisturbed terrain allows us to investigate the hydrologic characteristics while controlling for variable geology and soil types. We believe this to be the first successful bNMR dataset in permafrost of its type.





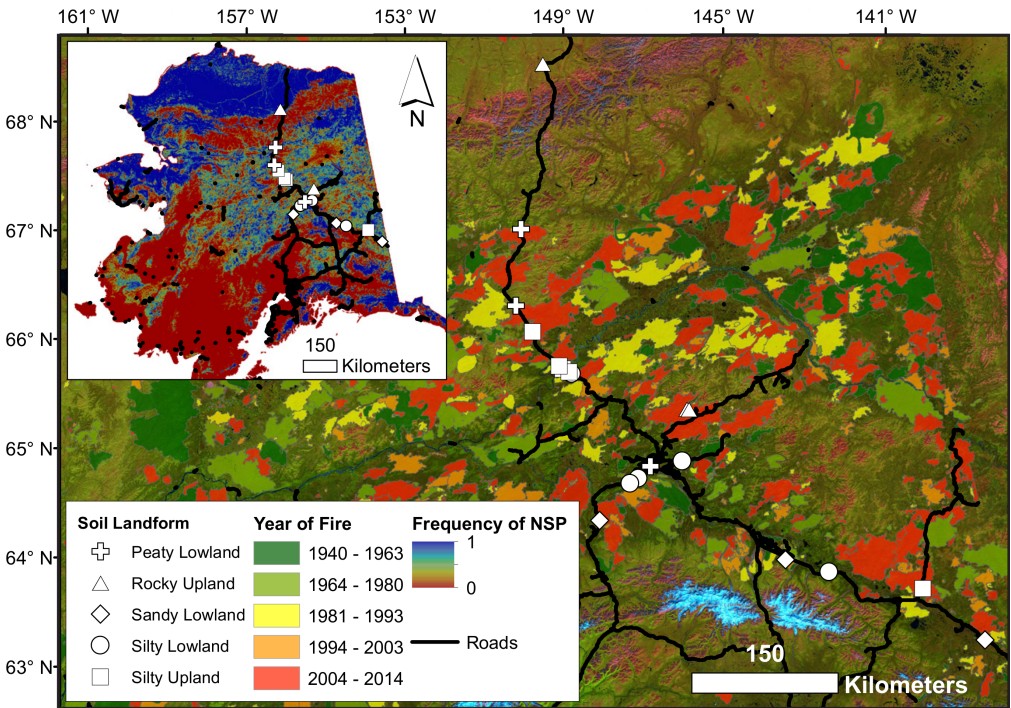

**Figure 1.** Field study area with wildfire boundaries since 1940 (Fire management records available from the Alaska Interagency Coordination Center at http://www.fire.ak.blm.gov). Study sites marked with white symbols indicating predominant soil landform. *Inset:* State map of Alaska with prediction of near surface permafrost (Pastick et al., 2015) and site locations.

## 1.1 Field Area

This study contains data primarily from the boreal forests of Alaska, though a few data points were acquired farther north in tundra regions (Fig 1). Each survey location was selected to span the boundary between burned landscape from a wildfire and unburned forest, at a variety of burn ages, terrains, and permafrost continuity (Pastick et al., 2015). In total, 37 boreholes were
5     sampled across 25 site locations, covering a transect of over 900 km.

## 2 Methods

At each field site, one or more boreholes were drilled along a transect and immediately logged with a borehole NMR probe. Electrical resistivity tomography (ERT) was also measured along the transect, along with data on soil type and depth as well as vegetation type. Borehole temperature data were acquired in a few selected locations. Transects were constructed to cross
10     from wildfire-burned terrain into unburned terrain. All data were acquired in late summer/early fall, at maximal seasonal thaw.



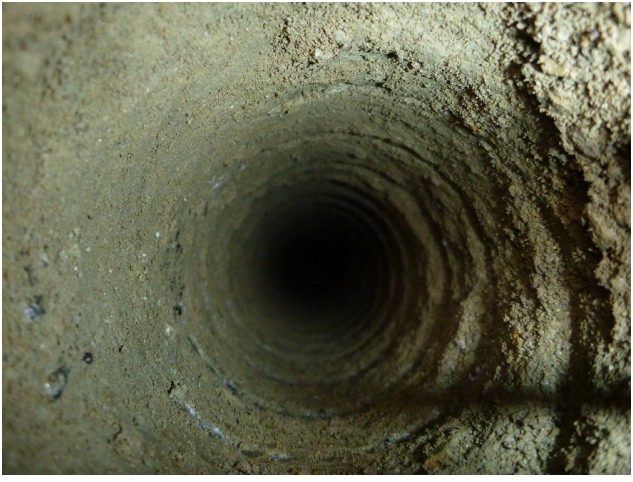

**Figure 2.** Photo of borehole after completion with gasoline-powered auger. Ice is visible along the walls where the soil was abraded, demonstrating the limited thermal effects of drilling.

## 2.1 Borehole Construction

Holes were drilled either manually with a soil auger (5.7 cm diameter) or using a gasoline powered auger with an ice bit (6.35 cm diameter). At five sites (site names prepended with 'CCHRC') at the Cold Climate Housing Research Center at the University of Fairbanks (www.cchrc.org), boreholes drilled with a rotary Geoprobe (Geoprobe Systems, Salina, Kansas, USA)

were logged between one hour and several days after drilling.

Both manual and gas augers successfully drilled through active layer and permafrost to depths of up to 2.75 meters while minimally disturbing the structure of the surrounding medium. Figure 2 demonstrates the efficacy of using a gasoline auger–ice is visible along the borehole walls showing the limited thermal interaction between the bit/drilling activity and the medium.

In some cases, heavy water saturation in the active layer caused the borehole to completely collapse above the permafrost

as soon as the auger was removed. Many holes were abandoned in these conditions; where measurements were made, we note that the estimation of the water saturation in the active layer may be high. This effect is most pronounced when the saturation is above 60 to 70%.

## 2.2 Data Acquisition

### 2.2.1 Borehole NMR

The nuclear magnetic resonance technique is the only geophysical tool directly sensitive to groundwater. When placed in a magnetic field, the proton in a hydrogen atom tends to precess with an axis parallel to that field. By generating an electromagnetic pulse or sequence of pulses at the Larmor frequency (which is a function of the individual atom's gyromagnetic ratio and the ambient magnetic field), the precession can be tipped. Once tipped, these precessing atoms–now also in phase–relax





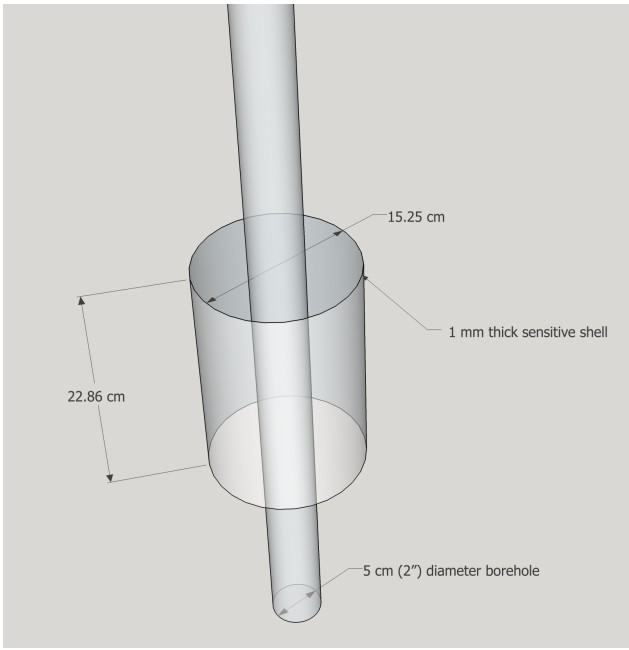

**Figure 3.** Schematic of the geometry of the sensitive zone. The instrument fits down a 5 cm borehole, and has a vertical sensitive zone of approximately 23 cm. The 1-2 mm thick sensitive zone is centered about a cylindrical shell approximately 15 cm in diameter.

back to their steady state, resulting in a measurable secondary electromagnetic field. The initial amplitude of this decaying secondary field is proportional to the total quantity of water within the volume of investigation. As water molecules move around a pore space (a function of Brownian motion), they have a probability of impacting a pore wall; when they do, they quickly 'relax' and contribute to the measured secondary field at a different decay rate. Therefore the measured signal is a function of

total volumetric water as well as the surface area to volume ratio of the pore (pore size). With ancillary information, hydraulic conductivity can also be estimated (e.g., Kenyon, 1997). These measurements can be made from borehole or surface systems. It may be helpful toward understanding to note that the physics of the NMR geophysical measurement are the same as that of a medical magnetic resonance imaging (MRI) scan.

Borehole data were acquired with a Dart NMR *in situ* soil moisture probe (Vista Clara, Inc., Mukilteo, Washington, USA).

The probe has a diameter of 4.4 cm with a sensitive zone consisting of a 1-2 mm thick cylindrical shell centred at the middle of the tool with a diameter of 15.3 cm (well outside the borehole disturbed zone) and a vertical extent of 22.9 cm (Fig. 3). The relatively light and compact nature of the instrument coupled with the portable augers allows for data acquisition in remote and rough terrain (Walsh et al., 2013).

Within each borehole, the instrument was held at a static depth for the duration of one record, then moved to acquire data

at the next desired depth. At each location, the instrument acquired two Carr-Purcell-Meiboom-Gill (GPMG) pulse datasets (Meiboom and Gill, 1958): a long (3 second) record and a "burst mode" with a 0.8 second record, which are combined to improve the signal-to-noise ratio in the first 0.8 seconds of the decay. The long record consisted of 374 echoes with 50 stacks





while the burst mode contained 62 echoes and a minimum of 200 stacks. The pulse length was 0.06 ms with an interpulse delay of 0.8 ms. Because of the background field defined by permanent magnets in the instrument, data were recorded at two fixed frequencies: 426.27 kHz and 478.271 kHz which further improves the signal-to-noise ratio of the resulting measurements. Before both field campaigns, the instrument was calibrated by the manufacturer (Walsh et al., 2013). Calibration was found to

5 be satisfactory through measurements of bulk water and air. A total record at one depth takes approximately 5 minutes to log, depending on number of stacks.

### 2.2.2 Other Data Acquisition

Coincident with the NMR data, surface direct current (DC) resistivity data were acquired along transects of varying length. This improved the interpretability and confidence of the permafrost depth measurements and revealed subsurface permafrost

structure. For this study, the DC data were used to cite optimal borehole locations and target depths.

In addition to geophysical data, depth to permafrost was measured with a permafrost probe (a long, thin metal spike driven into the soil). Care was taken to differentiate between seasonal frost and permafrost (both using the permafrost probe and through comparison with the resistivity models later); however seasonal frost may be represented in the frozen soils data. Organic layer thickness was measured along each geophysical transect.

Each location was classified into one of four soil types: silty, sandy, peaty, or rocky. Terrain was classified as upland or lowland. Clearly, any one field site could exhibit more than one soil type; these classifications reflect the predominant soil present.

In a few sites, temperature probes were installed. Flexible polyethylene (PEX) tubing was inserted as deeply as possible, then temperature measured with a thermistor after the tubing was allowed to thermally equilibrate.

## 2.3 Data Processing

### 2.3.1 Borehole NMR

The NMR data were processed using commercial software (Vista Clara, Inc.) to despike and stack the data and produce estimates of water content and pore size distribution. The data were first culled to remove records contaminated by spikes, then the two pulses at each frequency were combined to form one averaged time series, known as the $T_2$ or spin-echo decay. A

25 single $T_2$ decay was constructed for each depth at each borehole. The time series can be thought of as a linear combination of exponential decays, each with a single time constant ($T_2$ value), e.g.:

$$f(t) = \sum_i^n a_i e^{-t/T_{2i}} \qquad (1)$$

where $a_i$ is the amplitude of the exponential and $n$ is the number of $T_2$ bins (Brownstein and Tarr, 1979; Walsh et al., 2013). In bNMR surveys of this type, it is assumed that surface relaxivity dominates the decay (neglecting the long decay of bulk water





and strong magnetic gradient effects) and thus $T_2$ is inversely proportional to the pore geometry by:

$$\frac{1}{T_2} \approx \rho \frac{S}{V} \tag{2}$$

where $S$ is the surface area of the pore, $V$ is the volume, and $\rho$ is surface relaxivity: a property of the surface to cause relaxation of the fluid (Kleinberg, 1996). Thus Eq. 1 represents the decay from a distribution of pore geometries within a volume. For

relatively simple pore geometries, $S/V$ is inversely proportional to size and and so large pores result in long $T_2$ times.

Through regularized multiexponential fitting to the decay curve (Eq. 1), estimates of the total volumetric liquid water content and partial water contents as a function of pore size were constructed. Total volumetric water content is (with proper calibration) given by the initial amplitude of the multiexponential decay, while the estimates of volumetric water content as a function of pore size are derived from the integrated partial water contents as a function of $T_2$. Regularization was chosen through a manual

search and found to be near optimal at the default values.

Models of water content at each depth can be concatenated to produce a plot of water saturation as a function of both depth and $T_2$, which is a linear function of pore size. The conversion of $T_2$ to pore size was not performed due to insufficient calibration data; however relative relationships hold: the shorter the $T_2$, the smaller the effective pore size. Throughout this study, we refer to water as clay-bound ($T_2$ less than 3.16 ms), where water is bound to clay surfaces or otherwise in extremely

tiny pore spaces; capillary-bound ($T_2$ between 3.16 and 31.6 ms), where water is held by capillary forces in small pores; and mobile ($T_2$ greater than 31.6 ms). We note that these ranges are derived from oil-field NMR, and thus have limited application to the differing soils in this study. Moreover, the divisions are more of a gradient rather than a step-function. We do, however, find the descriptions illustrative and therefore useful, and these $T_2$ divisions likely serve as a 'lower bound' for the actual values. We refer the reader to Pallatt and Thornly (1990) for a full discussion on the divisions between bound, capillary, and

mobile water. The boundaries are plotted on the model results as dashed lines for reference.

## 3  Results

In this section, we show an example of the NMR interpretations for a field site with an observation in both burned and unburned terrain. We then generalize the results through a statistical analysis of the total water content between frozen and unfrozen soils, as well as the apparent pore size distribution.

A single NMR dataset corresponds to the NMR signal at one depth in one borehole, and is generally plotted as a stacked time-series decay; for the CPMG experiment, the time series represents the CPMG echo train envelope. Figure 4 shows the CPMG echo train envelope curve ($T_2$ decay) for an example field site, AK204-ROAD, which contains data from both burned (Fig. 4a) and unburned 4b) areas. Each plot is from the same soil type at the same depth below ground surface. The best-fitting multiexponential decay is superimposed (Eq. 1). Note the long, smooth decay exhibited by the unfrozen, saturated soil (Fig.

4a) in contrast to the rapid decay of tightly bound water exhibited by the frozen soil (Fig. 4b).

Figure 5 shows the calculated water contents from site AK204-ROAD as a function of pore size for both burned areas where near-surface permafrost had thawed (Fig. 5a) and unburned areas where permafrost remained intact (Fig. 5b) . The left panel



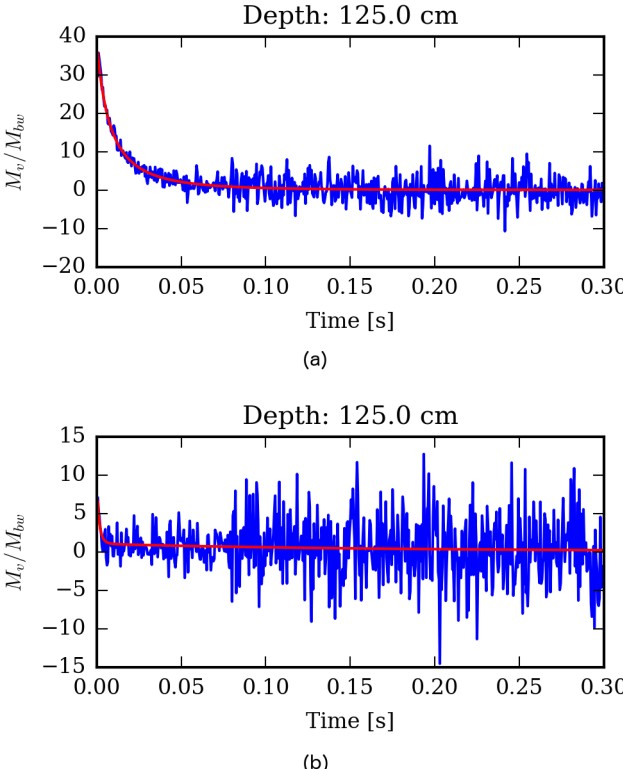

**Figure 4.** Example CPMG echo train envelope curves from AK204-ROAD at 125 cm depth for unfrozen (a) and frozen (b) soils. Soil type is consistent across the plots. Vertical axis is in NMR response normalized by the calibration factor and is thus in units of percent water content. Red curve indicates the best multiexponential decay fit.

shows the relative amounts of bound, capillary-bound, and mobile liquid water as a function of depth; the sum of those yields the total water content. The right panel shows the calculated $T_2$ distribution (linearly related to pore size) as a function of depth. Integrating across a depth from 0 to the bound limit yields the estimate of bound water. As in nature, the pore sizes generally demonstrate a log-normal distribution.

5    AK207-POLY is a tundra location dominated by ice wedge polygons. The computed model is shown in Fig. 6. This site was particularly interesting as the borehole was drilled into an ice wedge. The near surface was oversaturated with standing water (due to the ice wedge acting as an aquitard) creating a slurry of fine silt. Beneath the thin active layer, the borehole was drilled into massive ice.

For statistical analysis, results from each depth at each observation location were divided into frozen or unfrozen, and one

10    of each of the soil types at each measurement (Table 1). Results should not be treated as a random sampling of terrains; there is significant sampling bias present. It is difficult or impossible, for example, to drill through rocky terrain, and thus those locations are undersampled.





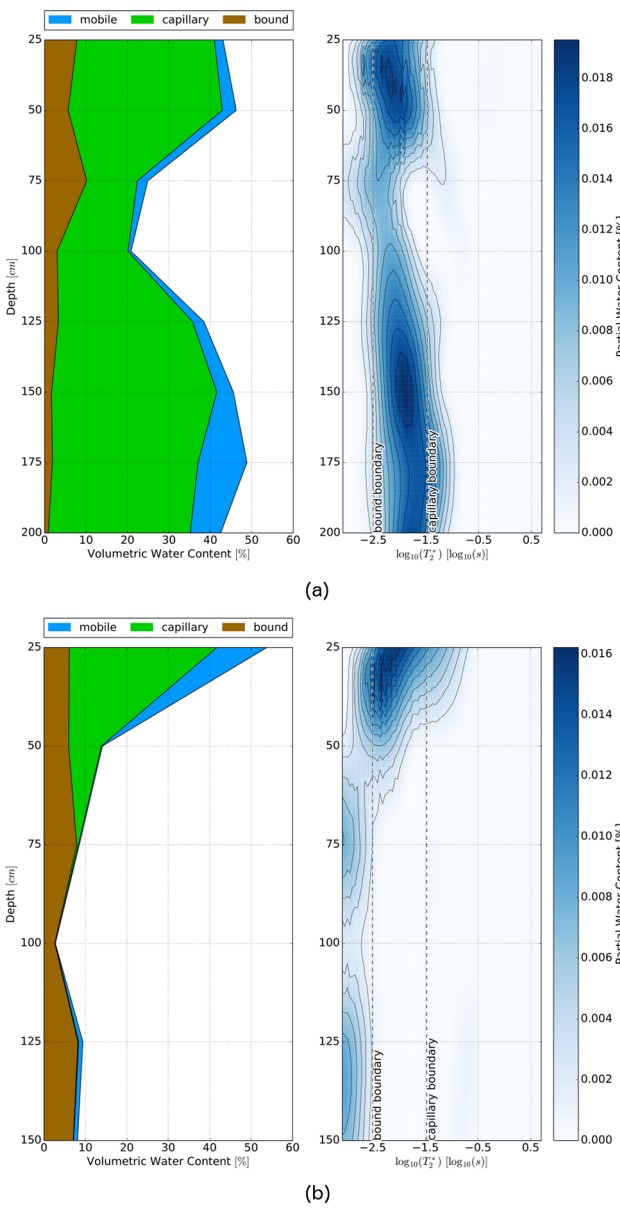

**Figure 5.** Model constructed for two boreholes at AK204-ROAD. (a) Unfrozen model. (b) Frozen model. Data acquired at depths enumerated in each plot. Note the difference in vertical scale due to different borehole depths.



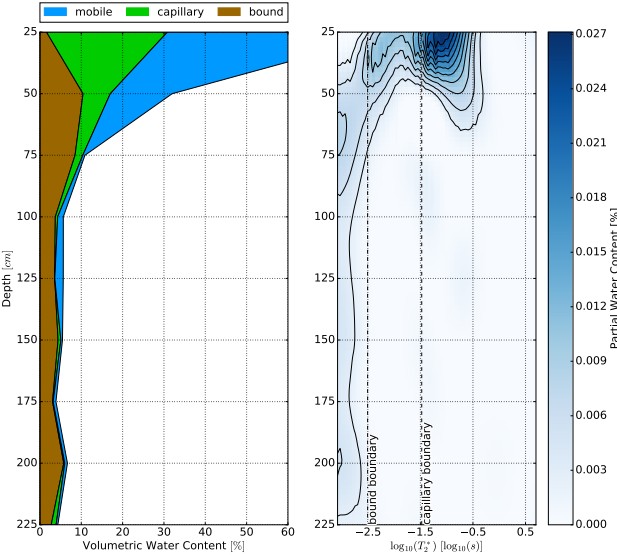

**Figure 6.** Model constructed from borehole measurements in an ice wedge at AK207-POLY. Data acquired at depths enumerated on the plot.

**Table 1.** Number of observations for each soil type classified into frozen or unfrozen.

|       | Frozen | Unfrozen |
|-------|--------|----------|
| Silty | 28     | 55       |
| Sandy | 2      | 28       |
| Peaty | 21     | 2        |
| Rocky | 6      | 3        |

## 3.1 Statistical Analysis

Based on active layer thickness from permafrost probe measurements and the DC resistivity inversions along each transect, the recovered model parameters were sorted into frozen and unfrozen categories. We note the limited presence of seasonal frost in these datasets. Figure 7 shows a composite histogram for the total water contents of frozen and unfrozen soil. Figure 8 shows the same information displayed using a box-and-whisker plot. The box of the box-and-whisker plot represents the interquartile range (IQR), or the central 50% of the data, with the median shown as a horizontal line. The whiskers are defined as the extremes of the dataset, excluding those data points outside of 1.5 times the IQR, or 99.3% of the data, equivalent to $2.698\sigma$ (standard deviation). Outliers are plotted as pluses.

The soils absent of permafrost show a mean water saturation of 38% ($n = 99$) with a standard deviation of 14%. Samples with permafrost present ($n = 90$) have a mean water saturation of 8% with a standard deviation of 6%. Most of the large deviation in saturation is a consequence of the large variability in sandy soils.





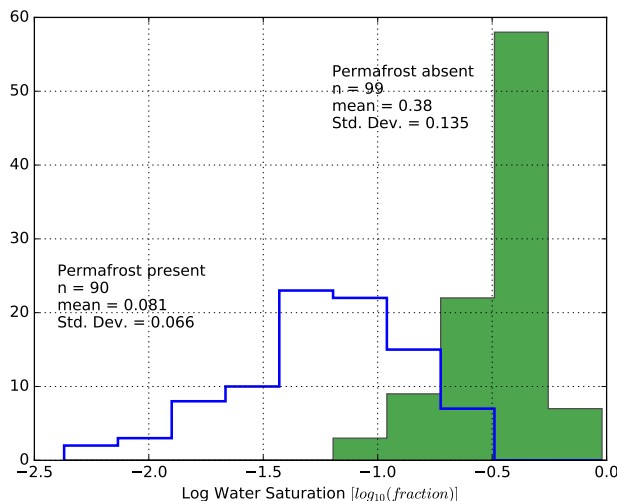

**Figure 7.** Histogram of bNMR results showing counts of total water content for data acquired in frozen zones (blue line) and unfrozen zones (green shaded). Measured water contents greater than 60% have been culled. Results displayed on a logarithmic water saturation scale for ease of viewing the distributions.

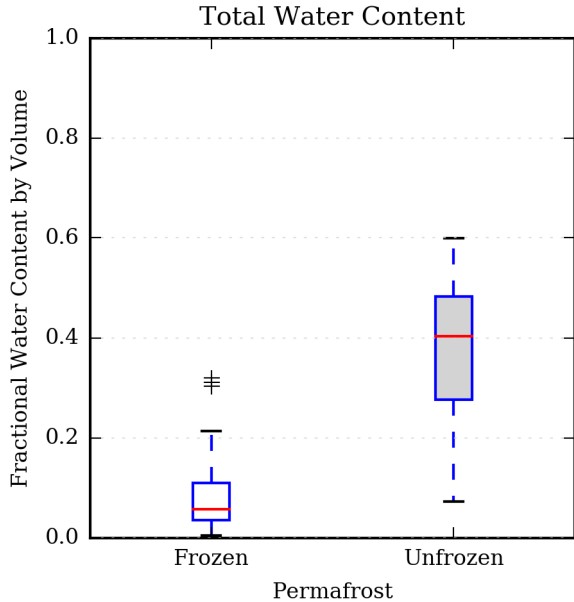

**Figure 8.** box-and-whisker plot of total volumetric water content as a function of permafrost presence or absence (active layer measurements are included in the 'absent' category). The 'box' represents the interquartile range (IQR, or 50% of the data) with the 'whiskers' defined as 1.5 times the IQR. Outliers plotted as 'pluses'; the line within the IQR represents the median.





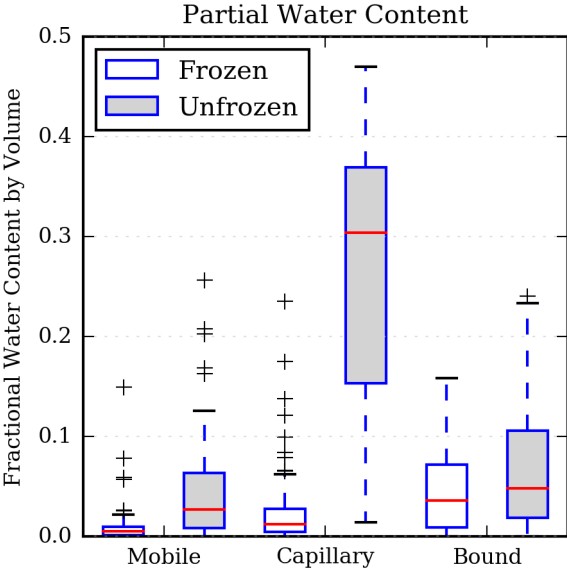

**Figure 9.** box-and-whisker plot of partial volumetric water contents as a function of permafrost presence or absence. Elements on the left of each pair correspond to the presence of permafrost; those on the right correspond to the absence of permafrost.

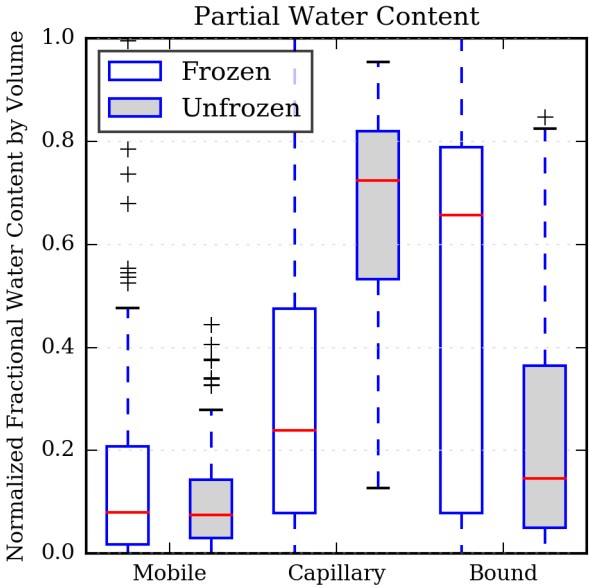

**Figure 10.** box-and-whisker plot of partial volumetric water contents as a function of permafrost presence or absence normalized by total water content. Elements on the left of each pair correspond to the presence of permafrost; those on the right correspond to the absence of permafrost.





Figure 9 shows how the aggregate bound, capillary-bound, and mobile water contents change as a function of freezing, assuming similar soil types between active layer and permafrost within each borehole. The samples of frozen and thawed measurements for clay, capillary, and mobile water were log-transformed (assuming a log-normal distribution) and each category was tested to see if the sample means between thawed and frozen were statistically different. Both a Student's T-test (assuming log-normality) and a Kolmogorov-Smirnov statistic (non-parametric) were used; both yielded equivalent results assuming the null hypothesis of equivalent distributions. As expected, water that is already bound to clay or other minerals with large surface area within the pore space is difficult to freeze and thus the values do not change significantly between frozen and thawed soils ($p > 0.01$). Conversely, the mobile water content is significantly reduced from thawed to frozen ($p < 0.01$), as is the capillary-bound water ($p < 0.01$), consistent with results from previous studies.

Figure 10 shows a box-and-whisker plot normalized by the total water content. In general, the $T_2$ values of the liquid water bound to pore surfaces remains largely unchanged, and thus the relative contribution of bound water to the total water is increased. The ratio of mobile water to total water remains relatively constant, however. This counterintuitive proportionality confirms the conclusions of previous laboratory analyses and modeling: ice nucleates in the center of pores, and liquid water remains in contact with rock or soil surfaces at the pore exterior (Anderson, 1967; Churaev et al., 1993; Kleinberg and Griffin, 2005). Based on these results, we concur with Kleinberg and Griffin (2005) that the water-soil NMR relaxation dominates the water-ice relaxation (thus the surface relaxivity for soil or rock as given in Eq. 2 is much larger than that of ice). $T_2$ values are proportional to the surface area-to-volume ratio of the pore (rather than absolute pore volume); since the water-ice interface does not largely contribute to the surface area in terms of relaxation, the $T_2$ values are consistently reduced with increasing frozen water content.

## 4    Soil Freezing Simulations

We constructed a soil freezing simulation to test both the null hypothesis–that ice nucleates in the center–and an alternative–that ice coats pore walls and aggrades inwards. Understanding the distribution of the water and ice phases within the permafrost media has important hydrologic implications on fluid flow and thermal properties. Ice phase water contributes no measurable signal at the frequencies and field strengths encountered in bNMR. As such, interpretation of the data hinges on also understanding the missing ice signal. Numerical simulation provides a mechanism to explore possible distributions of ice and water phases within a pore matrix. The presence of ice will influence the signal, even though it is not directly contributing energy to the signal. One major source of this influence is that ice has a characteristically low surface relaxivity ($\rho$) value, and depending on its location within a pore, it could substantially alter the relaxation rate of spins within pores. For instance, a thin layer of ice covering the grain surface within a pore could mask the surface relaxivity of the grain, increasing the relaxation time but having a minimal impact on the amplitude of the data. Conversely, ice within a pore will not mask the grain surface relaxivity and will therefore predominantly impact the signal amplitude. For this reason, empirically determined cutoff values for mobile-, capillary-, and clay-bound water used in the oil and gas industry–and more recently applied in near-surface hydrology–may




not be appropriate for interpreting these data. To better understand the controlling mechanism for signal decay, random walk simulations were performed using a suitable proxy model, with AK204-ROAD as a reference dataset.

A pore matrix model based on a thresholded $\mu$CT image of a silica gel was used as the starting point for an input model (Blunt et al., 2014; Dong and Blunt, 2009). A (50×50×50) voxel subset was extracted with a porosity of 57%, in order to correspond to the observed data, which showed early times intensities greater than 50% NMR water. The actual image was taken at a resolution of 3.85 $\mu$m; the resolution was scaled down to 2.2 $\mu$m for simulations such that the grain matrix corresponded to silt grain sizes (3.9–62.5 $\mu$m).

Random walk simulation code developed by the Imperial College London (Talabi et al., 2009) was used to perform synthetic NMR simulations on the developed model. The open source simulation code was modified to accommodate two different surface relaxivity values: for ice and silt portions of the grain matrix. Inputs to the simulations were chosen to approximate the Dart instrument, and field data from AK204-ROAD were used to calibrate simulation inputs.

To explore the possible distribution of water phases within the matrix a simplified conceptual freezing model was then developed, with $\rho_{grain} = 9.5 \times 10^{-2}$ m/s and $\rho_{ice} = 2.0 \times 10^{-5}$ m/s. The value of $\rho_{grain}$ was calibrated in order to fit the reference bNMR data, and the relaxivity of ice was set to be several orders of magnitude lower. Freezing was assumed to nucleate from pore centers, away from throats or pore walls. A suite of progressively more frozen models with ice propagating from the pore centers towards the walls was then developed to simulate stages of freezing/thawing. Figure 11 shows a cut-away for three stages of the freezing model.

Fig 12 shows the predicted and measured NMR response for freezing models corresponding to 25, 50, and 75 cm depths at AK204-ROAD. Even though the model is simplified, the derived freezing model well predicts the NMR response for frozen silts. The experiment was conducted in reverse (available as supplementary material), where ice nucleated on the surface of the pores and aggraded inwards; the simulated data were completely unable to fit the observed data. We therefore posit that the most likely distribution of phases is that of ice nucleating from the centers of pores.

Implications of this include higher permeability than would be observed if the opposite were true. Additionally, for this reason, the NMR response of liquid phase water in partially frozen soils will be dominated by the sediment interactions. This finding supports the applicability of existing cutoff values for interpretation of pore sizes and water content distribution in permafrost media.

## 5 Discussion

Frozen water has an exceptionally short NMR decay time–the hydrogen (being bound in a crystal lattice) dephases almost immediately, well within the 'dead time' of the instrument (between transmitter shut-off and receiver turn-on). Thus Figs. 7 and 8 indicate only liquid water. The results are as expected–when data are aggregated over all soil types, there is a clear separation in the histogram between liquid water present in unfrozen and frozen soils. This separation is not complete, however, because warm permafrost contains significant liquid water, even in the larger pores.





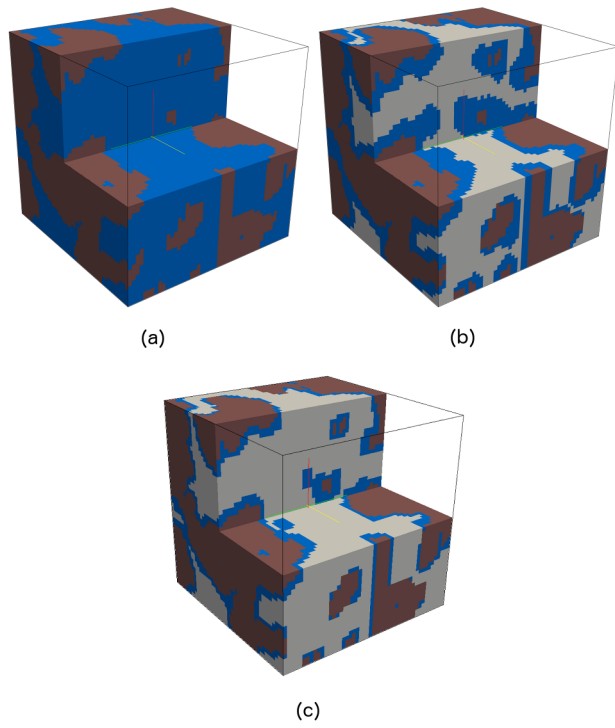

**Figure 11.** Soil freezing models for three progressive stages of freezing in pore spaces. Brown represents the silt, blue the liquid water, and white represents ice.

Characteristic freezing curves constructed by Williams and Smith (1989) indicate that clay-bound water can remain 20% liquid down to -5°C. Measurements by Romanovsky and Osterkamp (2000) show between approximately 2% and 9% unfrozen water by volume (at -5°C) depending on sample location, including a somewhat spatially coincident measurement at AK212-LTER. Comparison of results between the two studies (known as Bonanza Creek in the comparison study) indicate

5   the permafrost measured at 1 m depth had a calculated temperature of between -1°C and -2°C at the time of the bNMR measurement.

The consistency between borehole and laboratory observations seen in both the statistical analyses of field data as well as the freezing simulations provides good evidence that studying the NMR response of core is, in fact, a good proxy for *in situ* measurements. Thus bNMR and core analyses are complementary; bNMR can provide good spatial coverage and access at low

10  cost while core analyses can provide further detail and calibration between $T_2$ and pore size.

Though not the main focus of this study, it may be of interest to note the distribution of frozen/unfrozen soil as a function of soil type (Table 1). Though the choice of soil type exhibits significant sampling bias, the relative abundances of frozen versus unfrozen soil within each bin can be investigated. Peat is a phenomenal insulator, and therefore there are few sampled unfrozen





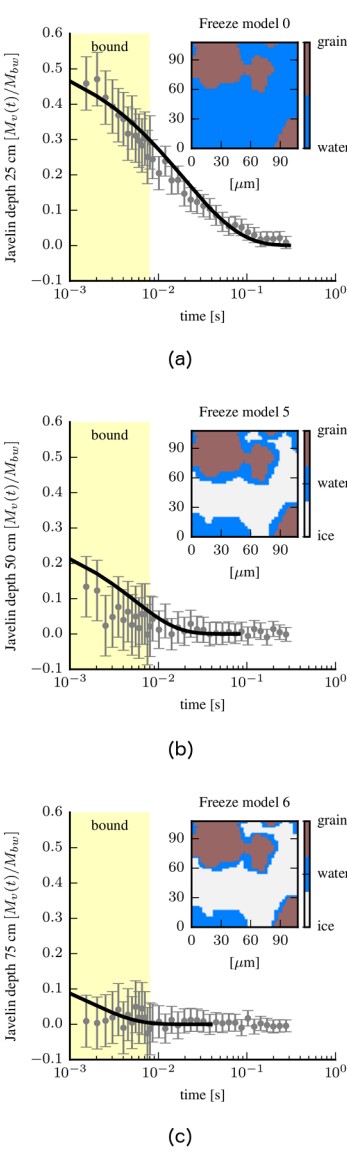

**Figure 12.** Observed (points) and predicted (line) data for progressively frozen models, corresponding to progressive depths in AK204-ROAD.





peaty points. Conversely, not only does sandy soil have a higher bulk thermal conductivity (Konavalov and Romain, 1973) and thus thicker active layer, but it is also more difficult to retain water in the near surface due to higher permeability.

## 5.1 Error Analysis

Total water content is directly given by the initial amplitude of the decay curve. Analysis of the stacked data yields error
estimates of the total volumetric water content between 2 and 10% depending on location.

The error in $T_2$ estimates is less straightforward. Inspection of Fig. 4 yields an obvious dependence of the error level on the amount of volumetric water. Low amplitudes and rapid decays result in fewer measurements in the time series available for an exponential fit, and the process becomes unstable. This instability can be controlled with regularization, but still contributes fundamental uncertainty in the final model. Techniques such as Bayesian analysis may help elucidate the contributions of the
short decays to uncertainty.

## 6 Conclusions

Borehole nuclear magnetic resonance is a rapid and reliable way to characterize the hydrologic properties of near-surface permafrost and active layer water. Boreholes constructed with hand or small gasoline-powered augers as well as larger Geoprobes can be logged with minimal concern for disruption of permafrost. Borehole NMR successfully measures the significant liquid
water present in warm permafrost while allowing for detailed analysis of pore size distribution as a function of depth.

Both *in situ* NMR measurements and soil freezing models agree with the hypothesis that ice nucleates in the center of pores in soil matrices. Statistical analyses of the NMR data are consistent with a low-relaxivity ice aggrading from the interior. Numerical simulations demonstrate the sensitivity of borehole NMR to this mechanism of ice formation by modeling a central nucleation as well as a pore-wall aggradation; ice formation from the pore wall is unable to fit the observed data while central
nucleation matches well.

The relative ease by which these measurements can be made allows for large spatial areas to be investigated, helping to understand the effects of climate change and increased impacts of wildfires on near-surface permafrost. Permanent boreholes can allow for monitoring of permafrost conditions on a variety of timescales, from daily to decadal. When combined with other data, such as hydrologic, geologic, and geophysical, the NMR technique fills an important gap in understanding complex
changes in permafrost environments.

## 7 Code availability

Random walk code from Imperial College London available at http://www3.imperial.ac.uk/pls/portallive/docs/1/50289716. ZIP.



## 8 Data availability

All data are freely available and archived with Science Base (Minsley et al., 2016a), and conform to Federal data release standards.

*Competing interests.* The authors declare that they have no conflict of interest.



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





*Disclaimer.* Any use of trade, firm, or product names is for descriptive purposes only and does not imply endorsement by the U.S. Government.

*Acknowledgements.* We thank Amy Marsh for invaluable field help and joviality. We also thank Elliot Grunewald and Dave Walsh of Vista Clara, Inc., and Tony Wong of the University of Colorado for numerous helpful discussions. Research was funded by the U.S. Geological Survey (USGS) Land Change Science Program's LandCarbon project (Z. Zhu Project Chief). A portion of this work was performed under USGS contract G08PC91508. Additional funding was provided by the Changing Arctic Ecosystems Initiative of the USGS Ecosystems Mission Area through the Alaska Cooperative Fish and Wildlife Research Unit. The Bonanza Creek Long Term Ecological Research Program and Institute of Arctic Biology's Toolik Field Station provided in-kind support and access to their research sites. We thank Art Gelvin of the Cold Regions Research and Engineering Laboratory, Army Corps of Engineers, for access to the CCHRC borehole sites.