# Peer review of "In situ nuclear magnetic resonance response of permafrost and active layer soil in boreal and tundra ecosystems"

_The Cryosphere, 2016_

## Referee Comment (RC1) · S. Akagawa (Referee) · 27 Mar 2017

General Comments

This paper evaluates the validity of bNMR for the observation of the climate change, ecosystems and wild fire disturbances in permafrost regions through intensive works. The context of this paper is found rational as far as the applicability of Kleinberg (1996) is affirmative. However the conversion model from T2 to pore diameter (Kleinberg, 1996) is developed for sand stone, it is rather questionable that the scenario is applicable to the case of soil.

Specific Comments

[Figure]

Major materials in active layer and frozen body close to permafrost table in low land of permafrost regions are soils but not rocks. However the concept of the pore in soil is totally different from that of sandstone, which is the model of Eq.2. The equation is valid if independence of the each pore is able to be rationally assumed. However due to the following empirical and theoretical research evidences, the assumption mentioned above will not be acceptable in soil pores.

<Specific Surface Area>

For example, the specific surface area of sandstone distributes around a few m2/g whereas that of clay and soil distribute from a few to 100m2/g, as shown in Table 1 (Akagawa and Syouji, 2004). The specific surface area of fine soil is about a few to 50 times larger than that of sandstone.

Table 1. Specific surface area of soils observed with BET method (Akagawa and Syouji, 2004)

Regarding to unfrozen water content, it is well known experimentally (Williams, 1964) and theoretically (Kuroda, 1985) that the fine soil has considerable amount of unfrozen water in subzero temperature. Even the amount sharply decreases from 0 to -1 Deg.C , it still exist at -20 Deg.C.

<Thickness of Unfrozen Water>

Regarding to the distribution of unfrozen water, it is discussed with its thickness from the surface of clayey minerals but not discussed with pore diameter. Because the specific surface area of fine soils is so extensive and the clayey minerals are generally plate-like shape, unfrozen water is believed to be distributed right on the surface of clayey minerals. The thickness of unfrozen water ranges from 10 to 100 nm at -0.1 Deg.C and 5 to 40 nm at -1 Deg.C (Akagawa and Syouji, 2004), as shown in Figure 1. Therefore unfrozen water decreases by thinning its thickness as its temperature

decreases.

Figure 1. Temperature dependent unfrozen water thickness (Akagawa and Syouji, 2004)

In addition, pore ice nucleation and growth behavior was studied by Black and Tice (1990) and Akagawa and Syouji (2004) and concluded that ice growth behavior due to freezing is the same as air intrusion behavior due to drying, as shown in Figure 2 and 3.

Figure 2. Thermodynamic condition of water in soils (Akagawa, 2005)

Figure 3. Distribution of water in freezing and drying soil (Black and Tice, 1988)

As the result, one of the conclusion of this article "ice nucleates in the center of pores in soil matrices" should be correct.

<Mobility of unfrozen water>

Regarding to the mobility of unfrozen water, there are two old works that confirmed the mobility of unfrozen water through frost heave research. The temperature of ice lens nucleation and segregation, i.e. "Ts", of alluvial clayey soil, during soil freezing was confirmed to be took place at the negative temperature of -0.8 Deg.C (Akagawa, 1990), as shown in Figure 4. And Ts=-1.4 Deg.C has confirmed in welded tuff (Akagawa et.al., 1988). Therefore water supplying to the segregating ice lens was understood to be supplied from unfrozen portion to growing ice lens through "frozen fringe" of soil and tuff of which temperature was 0 to -0.8 Deg.C and 0 to -1.4 Deg.C, respectively.

Figure 4. Segregation temperature (Ts) observed in frost heave test (Akagawa, 1990)

In other words, water flows from unfrozen soil/tuff to growing ice lens in unfrozen state through unfrozen water film of which thickness is about a few to 20 nm. According to the understanding mentioned above, the terminology "mobile water" used in this article

will not be suitable for defining bulk water. In other words, both bulk water and unfrozen water are "mobile water" in the temperature range of positive to -0.8 Deg.C in soil and positive to -1.4 Deg.C in tuff.

< Relation between T2 and unfrozen water thickness>

In addition, a preliminary work (Akagawa, 2005) demonstrated that T2 value of unfrozen water layer becomes shorter less than 1ms as the thickness of unfrozen water becomes less than 10 nm, as shown in Figure 5.

Figure 5. Relation between T2 and unfrozen water thickness (Akagawa, 2005)

The empirical result mentioned above may be indicating that the unfrozen water mobility might vary with the distance from the surface of clay minerals and/or temperature of the unfrozen water as far as the frozen soil is concerned.

<Comments>

As the result, it is inferred that without utilizing the temperature information of the target strata, the reliability of the analysis conducted in this article might be questionable. In addition, since the relationship between thickness of unfrozen film water and T2 vary with soil type (Akagawa, 2005), as shown in Figure 5, some kind of calibration must be required for the each soil type. Therefore it is recommended to comment 1) the applicability of "Eq.2" to soil which has large specific surface area, and 2) the rationality of "Figure 6" of the article by comparing with the specimens sampled from the bore hole, if available.

Reference

Akagawa, S., Goto. S. and Saito, A.; Segregation freezing observed in welded tuff by open system frost heave test, Proceedings, 5th International Conference on Permafrost, 1030-1035, 1988.

Akagawa, S.; X-ray photography method for experimental studies of frozen fringe characteristics of freezing soil, U.S.A. Cold Regions Research and Engineering Laboratory, CRREL Report 90-5,1990.

Akagawa, S. and Syouji, H.: Relation between T2 of pulse NMR and unfrozen water thickness, Proceedings of Hokkaido Branch of Japanese Geotechnical Society, 75-78, Sapporo, 2004 (in Japanese).

Akagawa, S.: Water properties in silty and clayey soils, The 2nd International Workshop on Gas Hydrate Studies and Other Related Topics, CD distribution, Kitami, 2005.

Black, P.B. and Tice, A.R.: Comparison of soil freezing curve and soil water curve data for Windsor Sandy Loam, U.S.A. Cold Regions Research and Engineering Laboratory, CRREL Report 88-16,1988.

Kuroda, T.: Theoretical study of frost heaving – Kinetic process at water layer between ice lens and soil particles, Proceedings of 4th International Symposium on Ground Freezing, 39-45, Sapporo, 1985.

Willams, P.J.: Unfrozen water content of frozen soil and soil moisture suction, Geotechnique, 14. 3. 231-246, 1964.

Figure with Y-axis "Unfrozen Water Thickness (nm)" ranging 0 to 100, and X-axis "Temperature (℃)" with values +1, |−0.1|, |−1|, |−10|, |−100|.

Legend:
- Kibushi Clay
- Dotan Silt
- NSF Clay
- Fairbanks Silt
- Bangkok Clay
- Ariake Clay
- Yokohama Clay
- Higashi-shinagawa Clay
- Fairbanks Silt
- Copper River Clay
- Hanover Silt
- Calgary Silt
- Dearmoun

**Fig. 1.**

[Figure]

**Fig. 2.**

[Figure]

**Fig. 3.**

[Figure]

**Fig. 4.**

**Fig. 5.**

Table 1. Specific surface area of soils observed with BET method (Akagawa and Syouji, 2004)

| Soil Name | Specific Surface Area (m²/g) |
|---|---|
| Higashi-Shinagawa Clay | 100.9 |
| Bangkok Clay | 50.6 |
| Saga-Ariake Clay | 37.5 |
| Kibushi Clay | 35.9 |
| Copper River Clay | 33.9 |
| Yokohama Clay | 31.3 |
| NSF Clay | 6.2 |
| Dotan Silt | 21.4 |
| Calgary Silt | 11.3 |
| Dearmoun Silt | 8.9 |
| Fairbanks Silt | 4.3 |
| Hanover Silt | 7.4 |

**Fig. 6.**

[Figure]

---

## Referee Comment (RC2) · Anonymous Referee #2 · 2 May 2017

The manuscript under review describes a study of NMR-response of temperate permafrost measured in situ in shallow boreholes. The approach is very innovative and the study is performed in very profound manner. The study contains the following important findings:

- variable state of permafrost can be investigated by means of borehole-NMR. The amplitude and decay-rates indicate qualitatively the amount of free water and its relaxation at the pore walls.

- Conventional approaches of categorization of water quantities by threshold values are not valid for water filled and partly frozen water.

[Figure]

- Numerical modelling of NMR-responses using an approach with frozen water in the center of the pores qualitatively explains the observed NMR-signals.

To my opinion the manuscripts has the following shortcomings:

- Petrophysical models of NMR-response from hydrocarbon exploration base on the assumption of a combination of water in oil in the pore space. In this case, the water will always cover the pore wall due to its bipolar character and thus affinity to the negatively charged pore-wall. The oil volume will be isolated from the relaxing pore wall and thus show relatively long decay-times. For common pore-space geometries this water will be a thin film and is thus called capillary water. Only for large pores or low oil volumes this water will have sufficiently large volume-to-surface ratios to show longer relaxation times > 30ms → free water. The authors recognize this limitation. Nevertheless they refer to this classification throughout the manuscript, even though they show in the course of the paper to be invalid. Additionally, the relaxation spectra in figure 5 give no evidence of a multiexponential distribution indicating different classes of water. The classification by hydrocarbon threshold values is misleading. I suggest to eliminate from the interpretation of the recorded data.

- Water in pores at or below the freezing point will be exposed to the forces of the negatively charged pore-wall and the crystallization to ice. As the numerical model clearly shows a model with ice-covered pore-walls cannot explain the measured data, while an ice-filled pore center qualitatively fits. Nevertheless, a description of the model with a homogeneous center of crystalline ice and a film of fluid water at the relaxing pore-wall is somewhat over-simplified. At the local freezing point (probably below 0°C, due to the presence of the pore wall) water will most probably not form a homogeneous crystal, but more likely a slush of ice and water. While solid ice at low temperatures has no measurable NMR-signal, in temperate ice, intercrystalline water is present in quantities that generate measurable NMR-signals at long relaxation times. A slush of ice and water will generate similar signals as the ice-filled pore in the study. Thus the model may be used to qualitatively differentiate the two models of ice crystals in
the center vs. ice covered pore walls. For a more quantitative analysis of the NMR-responses, the model is not suitable.

To conclude:

+ Detection of NMR-signals in-situ in temperate permafrost is a major finding. My congratulations!

+ Numerical analysis of the distribution of ice throughout the pore space is consistent with observed data and expected results.

- The classification according to oil-industry standards, valid for oil-water mixtures, is misleading an should be eliminated.

- The applied model with solid ice-crystals in the pore center and fluid film at the pore wall qualitatively matches the situation of freezing water within pores, but is limited for quantitative analysis.

---

## Author Comment (AC1) · 30 May 2017

We thank the referee for the constructive and positive comments, and have consequently added material into the manuscript. Specific responses to comments listed below.

- *Petrophysical models of NMR-response from hydrocarbon exploration base on the assumption of a combination of water in oil in the pore space. In this case, the water will always cover the pore wall due to its bipolar character and thus affinity to the negatively charged pore-wall. The oil volume will be isolated from the relaxing pore wall and thus show relatively long decay-times. For common*

[Figure]

*pore-space geometries this water will be a thin film and is thus called capillary water. Only for large pores or low oil volumes this water will have sufficiently large volume-to-surface ratios to show longer relaxation times > 30ms → free water. The authors recognize this limitation. Nevertheless they refer to this classification throughout the manuscript, even though they show in the course of the paper to be invalid. Additionally, the relaxation spectra in figure 5 give no evidence of a multiexponential distribution indicating different classes of water. The classification by hydrocarbon threshold values is misleading. I suggest to eliminate from the interpretation of the recorded data.*

For the case of water-wet rocks the reviewer is correct but the NMR response in oil-wet rocks exhibit the opposite behaviour. We believe the reviewer may be mistaking using $T_2$ for (fluid typing) rather than its use as a single phase permeability indicator. In environmental applications the use of cutoff values as an interpretation tool differentiating bound and mobile water are common (e.g. Behroozmand et al., 2015 and Knight et al., 2016). We acknowledge that without additional calibration these actual cutoff values are somewhat arbitrary, but have proven to be a useful as approximate bounds. The distributions are reasonably insensitive to small perturbations of these cutoff values. We have added discussion and references to the manuscript further highlighting the limitation of uncalibrated cutoffs. It may be useful for future literature to avoid the term capillary-bound and instead develop a new term in an unconsolidated setting to avoid confusion.

Figure 5 shows a distribution of exponentials (roughly log-normal distribution of pore sizes around a mean), but indeed does not conclusively indicate two separate populations of pore sizes.

- *Water in pores at or below the freezing point will be exposed to the forces of the negatively charged pore-wall and the crystallization to ice. As the numerical model clearly shows a model with ice-covered pore-walls cannot explain the measured data, while an ice-filled pore center qualitatively fits. Nevertheless, a*

*description of the model with a homogeneous center of crystalline ice and a film of fluid water at the relaxing pore-wall is somewhat over-simplified. At the local freezing point (probably below 0C, due to the presence of the pore wall) water will most probably not form a homogeneous crystal, but more likely a slush of ice and water. While solid ice at low temperatures has no measurable NMR-signal, in temperate ice, intercrystalline water is present in quantities that generate measurable NMR-signals at long relaxation times. A slush of ice and water will generate similar signals as the ice-filled pore in the study. Thus the model may be used to qualitatively differentiate the two models of ice crystals in the center vs. ice covered pore walls. For a more quantitative analysis of the NMR-responses, the model is not suitable.*

The pore scale model is included as first-order approximation of the distribution of the water and ice phases, and is not intended to capture the dynamics comprehensively. In addition, clay grains are below the resolution of the model scale. Regarding the NMR signal of slush, it depends on the mobility of the spins within the pores. If the ice fully occludes access to grain wall, then very long $T_2$ times will be observed. It would also be possible for the slush to result in restricted diffusion within the pores, which additionally could enhance the $T_2$ times. In our data we do not observe this phenomenon of long $T_2$ as depth increases and freezing is anticipated to also become more complete. However, a future rigorous laboratory study using a similar instrument in controlled freezing conditions would be fascinating to attempt to observe the phenomenon. We have added commentary discussing the consequences of nucleating ice characteristics to the manuscript.

**References**

Behroozmand, A., Keating, K., and E. Auken, 2015, *A review of the principles and applications of the NMR technique for near-surface characterization*. *Surveys in Geophysics*, **36**(1), p.

27-85.

Knight, R., Walsh, D., Butler, J., et al., 2016, *NMR logging to estimate hydraulic conductivity in unconsolidated aquifers. Groundwater*, **54**(1), p. 104-114.

---

## Author Comment (AC2) · 30 May 2017

We thank Dr. Akagawa for his thorough and detailed comments. The additional citations and information provided will enrich the paper. Specific comments are below.

- Regarding the use of NMR in unconsolidated media several clarifications can be made that may have not been evident in the initial manuscript. While it is accurate that porous media NMR was originally developed by the oil and gas industry where aquifers are often consolidated sandstone, there exists a wealth of examples of applying the Schlumberger-Doll and Timur-Coates equations to derive reliable permeability estimates in unconsolidated porous media (e.g. Knight et

al. 2012, Legchenko et al. 2004, Behroozmand et al. 2015, and Parsekian et al. 2015).

The use of NMR to characterize multiphase systems whereby one phase does not give appreciable signal (also considering unsaturated systems as an analogue for the ice phase) is somewhat less developed (Song 2010, Parsekian et al. 2013, and Walsh et al. 2014).

Given the rich literature in this space, we feel quite confident in the applicability of NMR measurements in the given context. These references have been added to the manuscript.

- *For example, the specific surface area of sandstone distributes around a few m2/g whereas that of clay and soil distribute from a few to 100m2/g, as shown in Table 1 (Akagawa and Syouji, 2004). The specific surface area of fine soil is about a few to 50 times larger than that of sandstone.*

Regarding the surface area to volume concerns of soil types: NMR instrumentation has shown capable of providing reliable estimates in unconsolidated media including clay layers (Knight et al. 2016) as well as tight oil and gas reservoirs with similar partial and corresponding $(S/V)$ (Xiao et al. 2012); although in these situations calibration of the coefficients in the Timur-Coates and Schlumberger-Doll equations may be necessary (specifically with regards to absolute pore sizes and permeabilities). The need for calibration was a primary motivation for the pore-scale simulations which we included in the manuscript. It has also been demonstrated that the cutoff values differentiating bound, and mobile water are much less sensitive than the exponential factors in permeability estimation.

- *Regarding to unfrozen water content, it is well known experimentally (Williams, 1964) and theoretically (Kuroda, 1985) that the fine soil has considerable amount of unfrozen water in subzero temperature. Even the amount sharply decreases from 0 to -1 Deg.C , it still exist at -20 Deg.C.*

Indeed, this is consistent with our findings and well known; we have added the additional references. Our primary interest is in characterising critical permafrost near 0° C where understanding the dynamics between the phases is important in management and forecasting.

- *Regarding to the distribution of unfrozen water, it is discussed with its thickness from the surface of clayey minerals but not discussed with pore diameter. Because the specific surface area of fine soils is so extensive and the clayey minerals are generally plate-like shape, unfrozen water is believed to be distributed right on the surface of clayey minerals. The thickness of unfrozen water ranges from 10 to 100 nm at -0.1 Deg.C and 5 to 40 nm at -1 Deg.C (Akagawa and Syouji, 2004), as shown in Figure 1. Therefore unfrozen water decreases by thinning its thickness as its temperature decreases.*

Thank you for making this point, which will also benefit from clarification. While we do expect to see some clay-bound water in these media, the NMR instrumentation is fairly insensitive to this water ($T_2 < 3.6$ ms). As such we do not surmise to be detecting the whole extent of clay bound water within the media. You are correct that for very small pores the $T_2$ times will become vanishingly small and will not be detectable using the instrumentation we have. Please remember, the sediments are primarily silty in nature (e.g. Fig. 2 and Table 1)–which from scanning electron imaging (Moraes et al. 2015) can be reasonably approximated using sphere packs or similar–hence our choice of models for simulation. The pore scale simulation scales are on the order of $\mu$m rather than nm, and the clay grains are effectively below the resolving power of $\mu$CT imaging. The pore-scale simulations are consistent with liquid-phase water distributed primarily at the surface of the pore matrix.

Specifically regarding the applicability of equation 2, we find it completely applicable due to the combination of (1) the NMR instrument being largely insensitive to the clay-bound water (and thus the distorted S:V) combined with the lack of

sampled clay soils, and (2) that we do not assume anything beyond the surface area to volume ratio (i.e. we do not quantitatively define pore sizes and shapes). The relative size relationships within the soils under investigation hold.

**References**

Knight, R., Grunewald, E., Irons, T., et al., 2012, *Field experiment provides ground truth for surface nuclear magnetic resonance measurement. Geophysical Research Letters*, **39**(L03304).

Legchenko, A., Baltassat, JM, Bobachev, A., et al., 2004, *Magnetic resonance sounding applied to aquifer characterization. Groundwater*, **42**(3), p. 363-373.

Behroozmand, A., Keating, K., and E. Auken, 2015, *A review of the principles and applications of the NMR technique for near-surface characterization. Surveys in Geophysics*, **36**:1, p. 27-85.

Parsekian, A., Dlubac, K., Grunewald, E., et al., 2015, *Bootstrap calibration and uncertainty estimation of downhole NMR hydraulic conductivity estimates in an unconsolidated aquifer. Groundwater*, **53**(1), p. 111-121.

Song, Y., 2010, *Recent progress of nuclear magnetic resonance applications in sandstones and carbonate rocks. Vadose Zone J.*, **9**, p. 828-834.

Walsh, D., Grunewald, E., Turner, P., et al., 2014, *Surface NMR instrumentation and methods for detecting and characterizing water in the vadose zone. Near Surface Geophysics*, **12**, p. 103-111.

Knight, R., Walsh, D., Butler, J., et al., 2016, *NMR logging to estimate hydraulic conductivity in unconsolidated aquifers. Groundwater*, **54**(1), p. 104-114.

Xiao, L., Mao, Z., Wang, Z., and Y. Jin, 2012, *Application of NMR logs in tight gas reservoirs for formation evaluation: A case study of Sichuan basin in China. Journal of Petroleum Science and Engineering*, **81**, p. 182-195.

Moraes, M., Lemos, V., Moraes, D., et al., 2015, *Characterization and distribution of pyrogenic carbon in a fraction of archaeological black earth from Caxiuanã. Journal of the Brazilian Chemical Society*, **26**, p. 1664-1673.